# NASViT: Neural Architecture Search for Efficient Vision Transformers with Gradient Conflict-aware Supernet Training

**Chengyue Gong**[2][*], **Dilin Wang**[1], **Meng Li**[1], **Xinlei Chen**[1], **Zhicheng Yan**[1],
**Yuandong Tian**[1], **Qiang Liu**[2], **Vikas Chandra**[1]

[1] Meta Reality Labs     [2] University of Texas at Austin

{wdilin, meng.li, xinleic, zyan3, yuandong, vchandra}@fb.com, {cygong, lqiang}@cs.utexas.edu

## Abstract

Designing accurate and efficient vision transformers (ViTs) is an important but challenging task. Supernet-based one-shot neural architecture search (NAS) enables fast architecture optimization and has achieved state-of-the-art results on convolutional neural networks (CNNs). However, directly applying the supernet-based NAS to optimize ViTs leads to poor performance - even worse compared to training single ViTs. In this work, we observe that the poor performance is due to a *gradient conflict* issue: the gradients of different sub-networks conflict with that of the supernet more severely in ViTs than CNNs, which leads to early saturation in training and inferior convergence. To alleviate this issue, we propose a series of techniques, including a gradient projection algorithm, a switchable layer scaling design, and a simplified data augmentation and regularization training recipe. The proposed techniques significantly improve the convergence and the performance of all sub-networks. Our discovered hybrid ViT model family, dubbed NASViT, achieves top-1 accuracy from 78.2% to 81.8% on ImageNet from 200M to 800M FLOPs, and outperforms all the prior art CNNs and ViTs, including AlphaNet and LeViT. When transferred to semantic segmentation tasks, NASViTs also outperform previous backbones on both Cityscape and ADE20K datasets, achieving 73.2% and 37.9% mIoU with only 5G FLOPs, respectively. Code is available at `https://github.com/facebookresearch/NASViT`.

## 1 Introduction

Transformers have recently been applied to various vision tasks, including image classification (Liu et al., 2021; Dong et al., 2021), object detection (Carion et al., 2020; Zhu et al., 2020), semantic segmentation (Xie et al., 2021; Cheng et al., 2021), video understanding (Bertasius et al., 2021; Fan et al., 2021), etc. Vision transformers (ViTs) benefit from high model capacity, large receptive field, and grouping effect, etc (Dosovitskiy et al., 2020), and demonstrate superior performance compared to convolutional neural networks (CNNs) especially with the scaling of the model size and training data size. For example, CoAtNet (Dai et al., 2021) achieves 90.88% top-1 accuracy on Imagenet by scaling the model to 2586G FLOPs and pre-training the model on JFT-3B dataset (Sun et al., 2017).

Though promising in the high computation budget regime, the performance of ViTs is still inferior to that of the CNN counterparts on small- or medium-sized architectures, especially compared to CNN architectures that are highly optimized by neural architecture search (NAS), e.g., AlphaNet (Wang et al., 2021a), FBNetV3 (Dai et al., 2020), etc. For example, the initial DeiT-Tiny (Touvron et al., 2020) only achieves 72.2% top-1 accuracy with 1.2G FLOPs. The recently proposed LeViT (Graham et al., 2021) makes significant progress to achieve 76.6% top-1 accuracy with 305M FLOPs with convolution/transformer hybrid architectures and a 3x longer training schedule. In contrast, AlphaNet (Wang et al., 2021a) achieves 77.8% top-1 accuracy with only 203M FLOPs. The large accuracy gap illustrated above raises a natural question: *are transformer blocks that build large and dynamic receptive fields beneficial for small models?*

---

[*]Work done during an internship at Meta Reality Labs

To answer the question above, in this work, we target at developing a family of efficient ViTs with FLOPs ranging from 200M to 800M . A natural approach is to leverage NAS, which has achieved state-of-the-art (SOTA) accuracy-efficiency trade-off for CNNs (Wang et al., 2021a; Dai et al., 2020; Cai et al., 2019). The recently proposed supernet-based NAS, e.g., BigNAS (Yu et al., 2020a) and AlphaNet (Wang et al., 2021a), builds a weight-sharing graph including all the sub-networks in the architecture search space. A sandwich sampling rule with inplace knowledge distillation (KD) (Yu et al., 2018) is leveraged to simultaneously optimize the supernet and sub-networks for each mini-batch, which stabilizes the training and improves the training convergence.

To leverage the supernet-based NAS, we first modify the LeViT model to build the architecture search space for ViTs and then jointly optimize the model architectures and parameters following AlphaNet. However, we find that directly applying AlphaNet achieves poor performance on the ViT search space, even worse compared to training single ViTs. To understand the root cause of the poor performance, we examine the supernet training procedure and observe that the gradients of the supernet and the different sub-networks conflict with each other during the sandwich sampling, which makes the training loss saturates much more quickly for ViTs, thus leading to slow convergence.

To alleviate the issue of conflicting gradients, we propose three different techniques to improve the supernet training. Firstly, instead of directly adding the gradients from different sub-networks together, we find it beneficial to prioritize the training of the sub-networks over the supernet, as our main purpose is to build efficient sub-networks. We achieve this with a projection gradient algorithm which removes the component of the supernet gradient that is conflict with the sub-network gradient. Secondly, to alleviate the gradient conflicts among different sub-networks, we propose to augment each transformer layer with switchable channel-wise scaling layers. The weights of different scaling layers are not shared among different transformer blocks to reduce gradient conflicts. Thirdly, we propose to use a weak data augmentation scheme and reduce the regularization in training to decrease the optimization difficulty and hence reduce gradient conflicts.

Our proposed techniques significantly alleviate the gradient conflict issue and empirically improve the convergence of supernet training. Compared to the baseline supernet training algorithm in AlphaNet, we can improve the top-1 accuracy to 78.2% for the small model with 205M FLOPs and achieve 81.8% for the large model with 757M FLOPs. Meanwhile, the resulting model family, NASViT, outperforms all the SOTA CNN and ViT models across a wide range of computation constraints. NASViT also demonstrates good performance on downstream tasks. When transferring to semantic segmentation tasks, NASViT backbones outperform previous CNN and ViT backbones on both Cityscape and ADE20K datasets, achieving 73.2% and 37.9% mIoU with 5G FLOPs, respectively.

**Related Works** Recently, researchers have used supernet-based NAS to optimize the architecture for transformers. For example, HAT (Han et al., 2021) uses supernet for hardware-aware transformer optimization. HAT mainly focuses on NLP tasks and features a design space with heterogeneous transformer layers. AutoFormer (Chen et al., 2021a) and ViTAS (Su et al., 2021) leverages supernet-based NAS to optimize the ViT architecture. By searching the width, depth, K/Q/V dimension, MLP ratio, etc, better accuracy is achieved compared to the baseline DeiT models (Chen et al., 2021a). However, these works focus on large ViT models with more than 1G FLOPs and their accuracy is still inferior to the CNN backbones with similar compute, e.g., EfficientNet (Tan & Le, 2019). We refer readers to appendix for more discussions about related works.

## 2 NAS FOR EFFICIENT TRANSFORMERS

Our goal is to design efficient small- and medium-sized ViTs in the FLOPs regime from 200M to 800M. We build our search space inspired by the recently proposed LeViT (Graham et al., 2021). LeViT is a family of efficient models leveraging a hybrid architecture of convolutions and transformers. In LeViT, the convolutions are introduced to handle high resolution inputs thanks to their efficiency from local computation while the transformers are leveraged for lower resolution features to extract global information. We closely follow LeViT to build our search space; see Figure 1 for an overview.

**Search Space** We summarize the detailed search dimensions of our search space in Table 1. For each CNN block, we directly follow the design in AlphaNet (Wang et al., 2021a;b) and search for the optimal channel widths, block depths, expansion ratios and kernel sizes; for each transformer

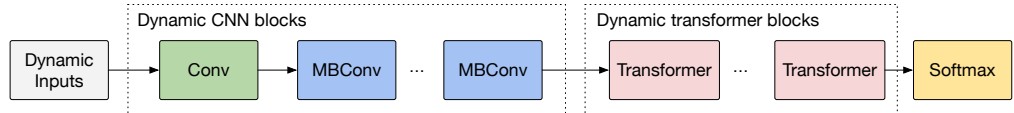

Figure 1: An illustration of our ViT search space. MBConv refers to inverted residual blocks (Sandler et al., 2018). All CNN and transformer blocks contain a stack of dynamic layers with searchable architecture configurations. Additionally, we also search for the input resolutions.

| Block | Width | Depth | Kernel size | Expansion ratio | SE | Stride | Number of Windows |
|---|---|---|---|---|---|---|---|
| Conv | {16, 24} | - | 3 | - | - | 2 | - |
| MBConv-1 | {16, 24} | {1,2} | {3, 5} | 1 | N | 1 | - |
| MBConv-2 | {24, 32} | {3, 4, 5} | {3, 5} | {4, 5, 6} | N | 2 | - |
| MBConv-3 | {32, 40} | {3, 4, 5, 6} | {3, 5} | {4, 5, 6} | Y | 2 | - |
| Transformer-4 | {64, 72} | {3, 4, 5, 6} | - | {1, 2} | - | 2 | 1 |
| Transformer-5 | {112, 120, 128} | {3, 4, 5, 6, 7, 8} | - | {1, 2} | - | 2 | 1 |
| Transformer-6 | {160, 168, 176, 184} | {3, 4, 5, 6, 7, 8} | - | {1, 2} | - | 1 | 1 |
| Transformer-7 | {208, 216, 224} | {3, 4, 5, 6} | - | {1, 2} | - | 2 | 1 |
| MBPool | {1792, 1984} | - | 1 | 6 | - | - | - |
| Input resolution | {192, 224, 256, 288} | | | | | | |

Table 1: An illustration of our search space. MBConv refers to the inverted residual block (Sandler et al., 2018). MBPool denotes the efficient last stage (Howard et al., 2019). SE represents the squeeze and excite layer (Hu et al., 2018). Transformer stands for the transformer blocks (Vaswani et al., 2017). For MBConv blocks, the expansion ratio refers to the expansion ratio of the depth-wise convolution layer. For transformer layers, it refers to the MLP expansion ratio. For each transformer block, we use $3 \times 3$ depth-wise convolution with stride 2 for down-sampling and the down-sampling layer is placed as the first layer for that block.

block, we search for the best number of windows, hidden feature dimensions (denoted as *Width* in Table 1)[1], depths and MLP expansion ratios. Compared to CNN blocks, one special search dimension for transformer blocks is the number of windows $k$. When the number of windows $k$ is greater than 1, we follow Swin transformer (Liu et al., 2021) and partition the input tokens into $k$ groups. We then compute the self-attention weights for each group separately to reduce computational cost. Standard global self-attention is a special case of $k = 1$. In this work, we only search the number of windows for the first transformer block, as the input resolutions to the other transformer blocks are already small after 4 times of down-sampling. Similar to the search range of AlphaNet, the smallest sub-network in our search space has 190M FLOPs and the largest sub-network has FLOPs of 1,881M. we refer the reader to Appendix B for more description of our search space.

**Naive supernet-based NAS fails to find accurate ViTs**   We first closely follow the previous best practices in AlphaNet (Wang et al., 2021a) for the supernet training. We train the supernet for 360 epochs on ImageNet (Deng et al., 2009). At each training step, we adopt the sandwich sampling rule (Yu et al., 2018) and sample four sub-networks: the smallest sub-network, the supernet (a.k.a. the largest sub-network), and two random sub-networks. All small sub-networks are supervised by the supernet with $\alpha$-divergence-based KD; see Algorithm 1 in Appendix C.1 for an overview of the supernet training procedure. Additionally, as our candidate networks contain transformer blocks, we further incorporate the best training recipe from LeViT (Graham et al., 2021) by replacing the SGD optimizer with Adam (Kingma & Ba, 2014) and leveraging an external pre-trained teacher model for the best accuracy. Specifically, we use the pre-trained teacher to supervise the supernet and still constrain all other small sub-networks to learn from the supernet. In this work, we always use an EfficientNet-B5 (Tan & Le, 2019) with 83.3% top-1 accuracy on ImageNet as the teacher to train our ViT supernet unless otherwise specified.

We plot the training curves of the smallest sub-network and the largest sub-network in Figure 2. We find both the smallest sub-network and the largest sub-network from our search space converge poorly compared to the CNN baseline. Specifically, the validation accuracy of both the smallest and the

---

[1]Hidden feature dimension equals the number of heads times the feature dimension of each head. In our search space, we fix the head dimension to be 8, and only searching for the number of heads.

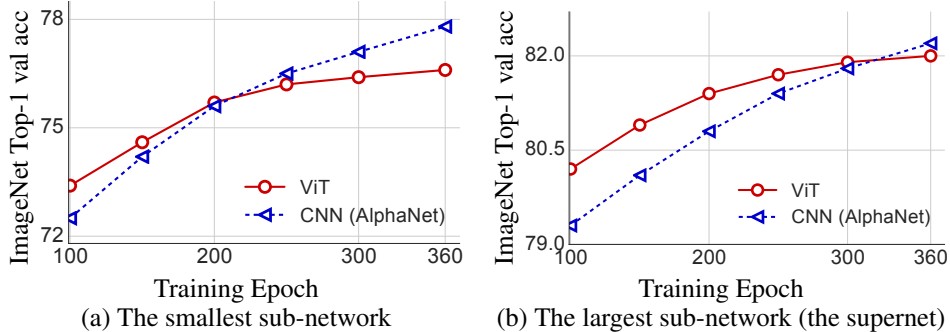

Figure 2: (a-b) show the training curves of the smallest sub-network and the largest sub-network (i.e., the supernet), respectively. Note that AlphaNet is trained without external teacher models.

| FLOPs (M) | 190 | 208 | 309 | 591 |
|---|---|---|---|---|
| Scratch | 77.2 | 77.5 | 79.1 | 80.4 |
| Supernet | 76.4 | 76.6 | 78.5 | 80.6 |

|  | AlphaNet | DeiT | LeViT |
|---|---|---|---|
| Smallest | 77.0 | 76.6 | 76.8 |
| Largest | 82.4 | 82.2 | 82.2 |

Table 2: ImageNet top-1 accuracy from sub-networks trained from scratch vs. results from sub-networks sampled from the supernet.

Table 3: ImageNet Top-1 accuracy from the smallest and the largest sub-network by using different training recipes.

largest sub-network is saturated at around the 250-th epoch, and the final accuracy is much worse than the CNN baselines. To understand the inferior model performance, we investigate the potential issues of our ViT supernet training from the following three directions.

**Investigation 1: Is our search space designed badly?**    We seek to understand if the performance gap is caused by a bad search space design. To verify, we randomly pick four sub-networks from the search space with computation cost ranging from 190M to 591M FLOPs. Then, we train these networks from scratch with the same data augmentation and regularization. As we can see from Table 2, the sub-networks trained from scratch outperform the sub-networks sampled from the supernet. Note that from previous works (e.g. Yu et al., 2020a), supernet often learns more accurate sub-networks compared to the training from scratch performance, by taking advantage of inplace knowledge KD and weight-sharing. Our observations in Table 2 indicate that the poor performance does not come from the search space but from the interference with the training of the supernet.

**Investigation 2: Are the training settings suitable for ViTs?**    Our default training settings from AlphaNet are originally optimized for CNNs only. Compared with AlphaNet, recent ViT methods, e.g., DeiT and LeViT, suggest to use stronger data augmentation schemes (e.g., a combination of CutMix (Yun et al., 2019), Mixup (Zhang et al., 2017), randaugment (Cubuk et al., 2020), random erasing (Zhong et al., 2020), and repeated augmentation) and stronger regularization (e.g., large weight decay, large drop path probability) for training. We evaluate the effectiveness of these ViT specific training recipes and summarize our findings in Table 3. As we can see from Table 3, DeiT- or LeViT-based training recipe produces even worse accuracy compared to the results from AlphaNet-based training.

**Investigation 3: Saturated supernet training due to gradient conflicts?**    Compared to the standard single network training, a major difference of supernet training is that multiple networks are sampled and trained at each step. We hypothesize that the training loss from the supernet and that from the sub-networks may yield conflicting gradients due to the heterogeneous and complex structures of networks, and the conflict gradients may consequently lead to slow convergence and undesirable performance.

To verify this hypothesis, we compute the cosine similarity between the gradients from the supernet and the averaged gradients from the sub-networks. A negative cosine similarity indicates the supernet and sub-networks produce conflict gradients and tend to update model parameters in opposite directions. To quantitatively examine the gradient conflict issue, we go through the entire ImageNet

| Epoch | 1st | 90th | 180th | 270th | 360th |
|---|---|---|---|---|---|
| AlphaNet | 27% | 20% | 21% | 24% | 28% |
| ViT | 36% | 27% | 27% | 32% | 34% |

Table 4: An estimation of negative cosine similarity ratio (gradient conflict ratio) between the supernet gradient and the averaged gradient of the sub-networks.

training set and calculate the percentage of negative cosine similarity between the gradients of supernet and sub-networks among all training images at a per layer granularity. The gradients are computed under the same data augmentation and regularization as the supernet training stage. For AlphaNet, we train the model using its official code [2]. As shown in Table 4, our ViT supernet suffers from more severe gradient conflicts compared to the CNN baseline. According to existing works in multi-task learning, large gradient conflict ratios may result in significant accuracy drop even for binary classification problems (see Figure 3 in Du et al. (2018) and Figure 4(b) in Yu et al. (2020b)). We hypothesize that the inferior performance of our ViT supernet is mainly caused by the large percentage of disagreements between the supernet gradients and the subnetworks gradients.

## 3 GRADIENT CONFLICT AWARE SUPERNET TRAINING

We propose to improve the ViT supernet training by addressing the gradient conflict issue between the supernet and the sub-networks from three aspects: 1) manually resolving the gradient conflict by projecting the supernet gradients to the normal vector of the sub-networks gradients; 2) introducing switchable scaling layers to the search space to give more optimization freedom for sub-networks; 3) reducing data augmentation and regularization to provide easier training signals.

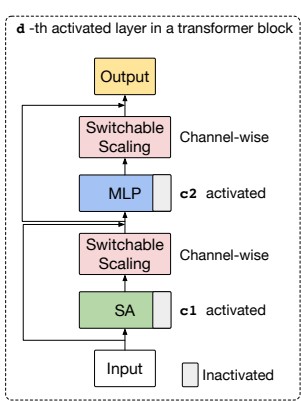

**Gradient projection to prioritize sub-networks update** Our first idea is to focus on training the sub-networks whenever gradients from the supernet and the gradients from the sub-networks conflicted with each other. As we are interested in the sub-networks from the 200M to 800M FLOPs range, we propose to prioritize the optimization of the sub-networks over the supernet when gradient conflicts are observed. Let $\nabla\ell_{sup}$ and $\nabla\ell_{sub}$ denote the gradients of the supernet and the sub-networks, respectively. To prioritize sub-networks training, we always project $\nabla\ell_{sup}$ to the normal vector of $\nabla\ell_{sub}$ to avoid gradient conflicts when the cosine similarity between $\nabla\ell_{sup}$ and $\nabla\ell_{sub}$ is negative. The overall accumulated gradient at each training iteration with projection can be written as follows,

Figure 3: A basic transformer layer with scaling. Activated components are the neurons selected in the forward path for one sub-network. 'c1' and 'c2' represent the number of channels activated in a self-attention layer and MLP, respectively.

$$g = \nabla\ell_{sub} + \mathrm{proj}(\nabla\ell_{sup}) \quad \text{with} \tag{1}$$

$$\mathrm{proj}(\nabla\ell_{sup}) = \begin{cases} \nabla\ell_{sup} & \text{if } \cos(\nabla\ell_{sup}, \nabla\ell_{sub}) \geq 0, \\ \nabla\ell_{sup} - \frac{\nabla\ell_{sup}^{\top}\nabla\ell_{sub}}{\|\nabla\ell_{sub}\|^2}\nabla\ell_{sub} & \text{otherwise.} \end{cases}$$

Note that $\cos(\nabla\ell_{sub}, \mathrm{proj}(\nabla\ell_{sup})) = 0$ if $\cos(\nabla\ell_{sup}, \nabla\ell_{sub}) < 0$, which ensures the gradient cosine similarity is non-negative. In sandwich sampling, since several sub-networks are sampled in each iteration, $\ell_{sub}$ is computed as the summation of the gradients from all sub-networks. Note similar ideas of gradient projection have also been explored in multi-tasks learning, see e.g., Yu et al. (2020b); Du et al. (2018); Real et al. (2019); Dery et al. (2021).

While the gradient projection in Eqn. (1) eliminates the gradient conflicts, it may lead to slow convergence as the resulting gradients are biased. Hence, we also propose the follow two techniques to reduce the gradient conflicts from a search space design and training strategy refinement perspective.

---

[2]https://github.com/facebookresearch/AlphaNet

**Switchable scaling layer**    Motivated by Slimmable NN (Yu et al., 2018), we introduce additional switchable scaling layers to allow sub-networks with different layer widths and depths to re-scale their features in a privatized way. Specifically, for each transformer layer, a switchable scaling layer is introduced at the output of the self-attention (SA) and the MLP, respectively, as shown in Figure 3. Assume $\boldsymbol{x}_{[c,d]} \in \mathbb{R}^c$ is a input feature of a scaling layer, with $c$ the feature dimension (i.e. the number of selected channels in the forward path) and $d$ the index of this layer in a transformer search block. The scaling layer transforms $\boldsymbol{x}_{[c,d]}$ as $\boldsymbol{w}_{[c,d]} \odot \boldsymbol{x}_{[c,d]}$. Here $\boldsymbol{w}_{[c,d]} \in \mathbb{R}^c$ are learnable parameters and $\odot$ denotes element-wise multiplication. For each transformer block (see Table 1), each different configuration of $[c,d]$ will specify a set of independent switchable scaling layers. Following CaiT (Touvron et al., 2021), we initialize all scaling factors $\boldsymbol{w}$ to a small value (e.g. $10^{-4}$) for fast convergence and stable training. Intuitively, the switchable scaling layers effectively increase the model capacity of sub-networks and give the sub-networks more optimization flexibility.

**Reduced data augmentation and regularization**    Furthermore, we observe that the supernet and the sub-networks are more likely to conflict with each other in the presence of stronger data augmentations and stronger regularization, e.g., large weight decay, large DropConnect (Wan et al., 2013). Hence, we simplify the AlphaNet training recipe and use a weaker data augmentation scheme - RandAugment (Cubuk et al., 2019) with both the number of augmentation transformations and the magnitude set to 1, and remove the regularization, e.g. DropConnect (Wan et al., 2013), dropout and weight decay, from the training; see Table 5 for a comparison.

| Method | Data augmentation | Weight decay | DropConnect | Dropout |
|---|---|---|---|---|
| AlphaNet | AutoAugment | $10^{-5}$ | 0.2 | 0.2 |
| Ours | RandAugment ($n=1, m=1$) | 0 | 0 | 0 |

Table 5: An illustration of our simplified training settings, where $n$ is the number of augmentation transformations and $m$ the number of magnitudes in RandAugment. A typical setting of RandAugment is $n=2$ and $m=9$ for training a single network; see Cubuk et al. (2020); Liu et al. (2021).

## 4    EXPERIMENTS

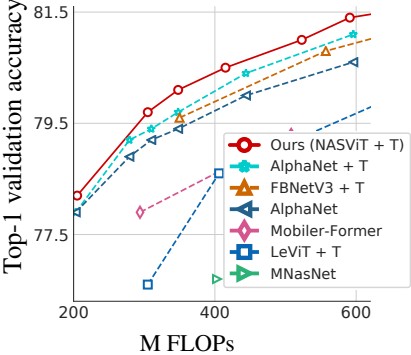

Figure 4: Comparison with prior-art CNNs and ViTs on ImageNet. Here "+*T*" indicates methods that are trained with external teacher models. Note that Mobile-Former (Chen et al., 2021c) and MNasNet (Tan et al., 2019) are trained without additional teacher models.

We first retrain our ViT supernet with our proposed gradient conflict reduction techniques on ImageNet (Deng et al., 2009); we then conduct an evolutionary search on a subset of the ImageNet training dataset to search the accuracy vs. FLOPs Pareto following (Wang et al., 2021b). We refer the reader to Appendix C.1 for more details. Note all the models are directly sampled from the supernet without retraining or finetuning.

We call our discovered as NASViT models and compare with state-of-the-art efficient CNNs and ViTs, including FBNetV3 (Dai et al., 2020), AlphaNet (Wang et al., 2021a), LeViT (Graham et al., 2021) and Segformer (Xie et al., 2021), on both image classification (e.g., ImageNet) and semantic segmentation benchmarks (e.g., Cityscapes and ADE20K).

### 4.1    IMAGENET

We compare our NASViT models with state-of-the-art NAS-based CNNs, including AlphaNet (Wang et al., 2021a) and FBNetV3 (Dai et al., 2020), and recently-proposed efficient ViTs, e.g., LeViTs.

**Settings**    Note that our ViT supernet is trained with a pretrained Efficient-B5 teacher model (83.3% top-1) model. For fair comparison, we retrain AlphaNet with the same teacher. For FBNet-V3 and

| Group | Method | M FLOPs | Top-1 accuracy (%) |
|---|---|---|---|
| 200-300 (M) | AlphaNet-A0 | 203 | 77.9 |
| | **NASViT-A0 (ours)** | **208** | **78.2** |
| 300-400 (M) | LeViT (Graham et al., 2021) | 300 | 76.6 |
| | **NASViT-A1 (ours)** | **309** | **79.7** |
| | AlphaNet-A2 | 317 | 79.4 |
| | FBNetV3 (Dai et al., 2020) | 357 | 79.6 |
| 400-500 (M) | LeViT | 406 | 78.6 |
| | **NASViT-A2 (ours)** | **421** | **80.5** |
| | AlphaNet-A4 | 444 | 80.4 |
| 500-600 (M) | **NASViT-A3 (ours)** | **528** | **81.0** |
| | FBNetV3 | 557 | 80.8 |
| | **NASViT-A4 (ours)** | **591** | **81.4** |
| | AlphaNet | 596 | 81.1 |
| 600 - 1000 (M) | LeViT | 658 | 80.0 |
| | **NASViT-A5 (ours)** | **757** | **81.8** |
| | FBNetV3 | 762 | 81.5 |
| > 1000 (M) | AutoFormer* (Chen et al., 2021a) | 1,300 | 74.7 |
| | PiT-XS (Heo et al., 2021) | 1,400 | 79.1 |
| | ViTAS-D* (Su et al., 2021) | 1,600 | 76.2 |
| | **NASViT (supernet) (ours)** | **1,881** | **82.9** |
| | CVT-13-NAS* (Wu et al., 2021) | 4,100 | 82.2 |
| | Swin-Tiny* (Liu et al., 2021) | 4,500 | 81.3 |
| | CVT-13* (Wu et al., 2021) | 4,500 | 81.6 |
| | T2T-ViT-14* (Yuan et al., 2021a) | 5,200 | 81.5 |
| | DeepViT (Zhou et al., 2021) | 6,200 | 82.3 |

Table 6: Comparison with prior art efficient CNNs and ViTs on ImageNet. The reported AlphaNet models are trained with an external teacher model. The "*" indicates that the ViTs are trained without external teacher models.

LeViT models, these models already use teachers with better performance than Efficient-B5 for training, and therefore we directly report their results following their papers. Specifically, FBNet-V3 use a RegNetY-32G with 84.5% top-1 and LeViT use a RegNetY-16G with 83.6% top-1 as the teacher model, respectively.

**Results** We summarize our results in both Table 6 and Figure 4. Our discovered NASViT models outperform all evaluated CNN and ViT baselines. Our models are the first models with transformers blocks that outperform state-of-the-art efficient CNNs with similar FLOPs on ImageNet. For example, with $< 600M$ FLOPs, our NASViT-A4 achieves 81.4% top-1 accuracy on ImageNet. As a reference point, a ResNet-50 model (4G FLOPs) achieves 81.5% top-1 accuracy by distilling from a BiT (Kolesnikov et al., 2020) teacher (87.5% top-1 accuracy) with 1200 epochs of training (Beyer et al., 2021).

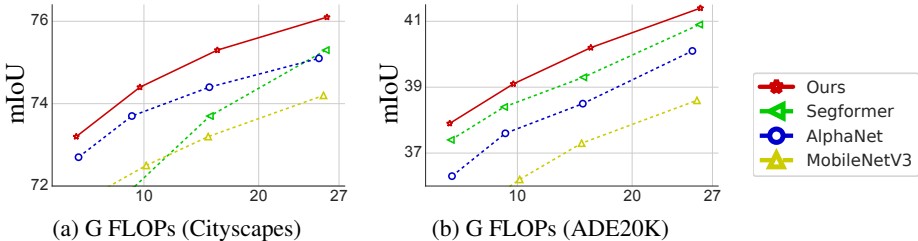

(a) G FLOPs (Cityscapes)                    (b) G FLOPs (ADE20K)

Figure 5: Results of our method and baselines on semantic segmentation. (a-b) show the results on the Cityscapes and ADE20K validation set, respectively.

## 4.2 SEMANTIC SEGMENTATION

We evaluate the transfer learning performance of our discovered NASViT models by fine-tuning them on downstream semantic segmentation tasks. In particular, we fine-tune NASViT-A1 to NASViT-A4 as backbones and we show that our NASViT models yield the best segmentation performance compared to the results from the previous efficient CNN backbones, e.g., AlphaNet

and MobileNetV3 (Howard et al., 2019), as well as the recently proposed transformer-based Seg-former (Xie et al., 2021).

**Settings**  We evaluate on two benchmark datasets, Cityscapes (Cordts et al., 2016) and ADE20K (Zhou et al., 2017). To handle large input resolutions efficiently, for each transformer block, we set the number of windows to be the input feature map size divided by 7 instead of using our searched settings on ImageNet. We use the recent proposed light-weight Segformer head (Xie et al., 2021) as the decoder head for all the backbones, to achieve a better accuracy efficiency trade-off.

**Results**  As shown in Figure 5, our models yield the best FLOPs vs. mIoU trade-offs. For example, our model achieves 76.1% mIoU and 41.4% mIOU with less than 30G FLOPs on the Cityscapes and the ADE20K validation set, respectively.

### 4.3  ABLATION STUDIES ON GRADIENT CONFLICT AWARE TRAINING

We conduct ablation studies on ImageNet to have a better understanding on the effectiveness of our proposed methods. We mainly study 1) how our techniques can mitigate the *gradient conflict* issues and improve the performance 2) whether CNN supernets can also benefit from our techniques. All the comparisons in this section are conducted on ImageNet.

**On the effectiveness of our gradient conflict aware training techniques**  As demonstrated in Table 7, both *weak data augmentation and regularization* and *switchable scaling layer* and can significantly reduce the gradient conflict ratios and in the meantime, improve the top-1 accuracy of both the smallest sub-network and the supernet. By further applying gradient projection to prioritize the sub-networks update (denoted by *Prioritize (sub)*), the performance of both the smallest and largest sub-network is boosted by around 0.3% on top-1 accuracy.

**Prioritizing the supernet update**  Instead of focusing on training the sub-networks, we retrain our ViT supernet and prioritize the supernet update by moving the $\mathrm{proj}(\cdot)$ term in Eqn. (1) to $\nabla \ell_{sub}$. As demonstrated in the last column of Table 7, this training strategy (denoted by *Prioritize (sup)*) leads to a slightly improved supernet while resulting in less competitive performance on the smallest sub-network.

|  | Baseline | Weak DA & Reg | Switchable scaling | Prioritize (sub) | Prioritize (sup) |
|---|---|---|---|---|---|
| Top-1 (smallest) | 76.6 | 77.4 | 77.6 | **78.1** | **77.9** |
| Top-1 (supernet) | 82.2 | 82.5 | 82.6 | **82.9** | **83.0** |
| Negative Cosine Similarity Ratio | 34% | 30% | 29% | 0% | 0% |

Table 7: Ablation study results on ImageNet. We show the top-1 validation accuracy of the smallest and largest sub-network, and the negative cosine similarity ratio for each case. Note that *switchable scaling layer* is applied on top of *Weak DA & Reg*; and *Prioritize (sub)* is applied on top of both *Weak DA & reg* and *switchable scaling layer*.

**Improving CNN-based supernets**  We verify the generalizability of our three techniques to the CNN supernets. In this setting, we applied all three techniques together to improve CNN-based supernets. We show in Table 8 that our method is especially helpful for AlphaNet trained with KL based KD (denoted by *AlphaNet (w/ KL)*). To further understand the large improvements on *AlphaNet (w/ KL)*, we follow ours studies in Table 4 and compute the gradient conflict ratio for *AlphaNet (w/ KL)* at epoch 1st, 90th, 180th, 270th, and 360th, and the corresponding gradient conflicts ratio is 25%, 18%, 24%, 28% and 31%, respectively. The gradient conflict issue is more severe for *AlphaNet (w/ $\alpha$-KL)* compared with AlphaNet trained with $\alpha$-divergence based KD (*AlphaNet (w/ $\alpha$-div)*). Our findings indicate that our techniques are not restricted to the ViT supernet training and might be beneficial for all supernets in which a large ratio of gradient conflicts presents.

### 4.4  ABLATION STUDIES ON SEARCH SPACE

In this part, we provide additional ablation studies to support some key design choices of our ViT search space. For all the studies, we use a 250M sub-network that is randomly sub-sampled from our search space for consistency.

| Architectures (M FLOPs) | A0 (203M) | A1 (279M) | A2 (317M) | A3 (357M) | A4 (444M) | A5 (491M) | A6 (709M) |
|---|---|---|---|---|---|---|---|
| AlphaNet (w/ KL) | 77.0 | 78.2 | 78.5 | 78.8 | 79.3 | 79.6 | 80.1 |
| **AlphaNet (w/ KL) + Ours** | **77.5** | **78.6** | **78.9** | **79.2** | **79.8** | **80.1** | **80.7** |
| AlphaNet (w/ $\alpha$-div) | 77.8 | 78.9 | 79.2 | 79.4 | 80.0 | 80.3 | 80.8 |
| **AlphaNet (w/ $\alpha$-div) + Ours** | 77.8 | 78.9 | 79.2 | 79.4 | 80.0 | **80.4** | **80.9** |

Table 8: Improving CNN-based supernets on ImageNet. AlphaNet (w/ KL) and AlphaNet (w/ $\alpha$-div) denote AlphaNets trained with KL and $\alpha$-divergence based knowledge distillation, respectively. A0 to A6 are the architectures reported in AlphaNet (Wang et al., 2021a). Note that the AlphaNet supernets here are trained without external teacher models.

**Global attention vs. local attention** In our search space (Table 1), we mainly use the global self-attention for the best representation learning capacity. However, with the more computationally efficient local and linear self-attention schemes, we would be able to use a slightly bigger model under similar FLOPs constraints with a sacrifice of global context modeling. To test this trade-off, we train the aforementioned model from scratch with different types of self-attention strategies. Specifically, in addition to the global self-attention, we further evaluate a number of local and linear self-attention mechanisms, including Swin (Liu et al., 2021), CSwin (Dong et al., 2021), VOLO (Yuan et al., 2021b) and LSH (Kitaev et al., 2020). We uniformly scale the width of the transformer blocks to ensure all models have similar compute FLOPs. In Table 9, we show that the standard global attention achieves the best accuracy compared to other faster local and linear self-attention methods. Additionally, from our evolutionary search results, we also notice that the sub-networks with all standard global self-attention layers often yield the best accuracy vs. FLOPs trade-offs.

| Global | Swin | CSwin | VOLO | LSH |
|---|---|---|---|---|
| 78.5 | 78.0 | 77.9 | 78.1 | 78.0 |

Table 9: ImageNet top-1 accuracy with different types of self-attention mechanisms.

**The placement of transformer blocks** Our ViT supernet has a convolution stem with 3 down-sampling strides. We further test the optimal choice of where to switch to transformer blocks. We uniformly scale the width of transformer layers of our baseline model to ensure similar FLOPs for different architecture designs. As shown in Table 10, we notice that our current design, a convolution stem with 3 strides, yields the best performance.

| #strides | 2 | 3 | 4 |
|---|---|---|---|
| Top-1 Accuracy | 78.1 | 78.5 | 78.4 |

Table 10: Ablation studies on where to switch to transformer blocks.

| Head Dimension | 8 | 16 | 32 |
|---|---|---|---|
| Top-1 Accuracy | 78.5 | 78.3 | 78.2 |

Table 11: Ablation studies on the impact of head dimension.

**Head Dimension** In previous ViT works (e.g Touvron et al., 2020; 2021; Zhou et al., 2021; Wu et al., 2021; Liu et al., 2021), the feature dimension of each self-attention head is usually set to be 32, 64, or larger. In this work, as shown in Table 11, we found a smaller head feature dimension (e.g., 8) yields better performance.

## 5 CONCLUSION

In this work, we identify one key issue of ViT supernet training that the supernet gradients and the sub-network gradients are likely to disagree with each other, and consequently leading to inferior NAS performance. We fix this gradient conflict issue by introducing a gradient projection method to prioritize the sub-networks update, designing switchable scaling layers to increase the model capacities of sub-networks and simplifying the training recipe to provide easier training signals. With our improved ViT supernet training techniques, our method finds a family of efficient models, called NASViT, that establishes a new state-of-the-art top-1 accuracy vs. FLOPs trade-offs on ImageNet. Our NASViT models are the first ViT variants that outperform prior-art efficient CNNs on the mobile FLOPs regime.

**Acknowledgements**   Chengyue Gong and Qiang Liu are supported in part by CAREER-1846421, SenSE-2037267, EAGER-2041327, and Office of Navy Research, and NSF AI Institute for Foundations of Machine Learning (IFML). We would like to thank the anonymous reviewers and the area chair for their thoughtful comments and efforts towards improving our manuscript.

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

# A   RELATED WORK

**ViTs**   ViT (Dosovitskiy et al., 2020) and its follow-ups (Wu et al., 2021; Liu et al., 2021; Zhou et al., 2021; Touvron et al., 2021) have been demonstrated to be an alternative choice to CNNs for challenging vision tasks, especially for image classification. In (Dosovitskiy et al., 2020), a large-scale ViT-Large model is trained on JFT-300 to obtain good performance. The follow-ups mainly focus on making the data size and model size smaller without loss of accuracy. A line of works introduce inductive bias or CNN layers to keep the good performance of ViTs while reducing the data and model sizes (e.g. Liu et al., 2021; Dong et al., 2021; Yuan et al., 2021b; Wu et al., 2021). For smaller data sizes, researchers successfully achieve good performance using ViTs without extra data. For example, VOLO (Yuan et al., 2021b) achieves 87.3% top-1 accuracy on ImageNet without extra data.

For smaller model sizes, however, ViTs have not achieved comparable results to efficient CNNS smaller than 1G FLOPs, even if additional CNN layers are introduced (Graham et al., 2021). Dynamic ViTs (Rao et al., 2021; Chen et al., 2021b) propose to dynamically filter the tokens to reduce the computation cost, and the efficiency is not comparable to efficient CNNs. LeViT (Graham et al., 2021), Xiao et al. (2021) and PiT (Heo et al., 2021) processes the high resolution inputs with early convolution layers or spatial-aware layers and also adopt more efficient self-attention designs. Mobile-former (Chen et al., 2021c) proposes a two-branch neural network: one is efficient CNN layers and the other is transformer layers with a small number of tokens (e.g. 6, 8).

**NAS**   NAS is a powerful tool for automating efficient neural architecture design. It often targets at searching for the best model in a search space under given efficiency-related constraints. Earlier NAS solutions often build on reinforcement learning (e.g. Zoph & Le, 2016; Zoph et al., 2018; Howard et al., 2019) and evolutionary algorithms (e.g. Real et al., 2019; 2017; Wan et al., 2020). More NAS practices have made the search more efficient through weight-sharing and search architectures with gradient-based methods (e.g. Liu et al., 2018; Pham et al., 2018; Stamoulis et al., 2019). This helps alleviate the heavy computational burden of training all candidate networks from scratch and accelerates the NAS process significantly. and researchers work on how to rank the model performance more accurately (Dong & Yang, 2020). Recently, training a large supernet without retraining candidate sub-networks with inplace KD is shown to be an effective mechanism that significantly improves the supernet performance (e.g. Yu et al., 2020a; Wang et al., 2021b;a). In addition to inplace KD, various of KD variants have also been investigated in the literature. For example, Peng et al. (2020) proposes to search a prioritized path as the teacher; Li et al. (2020) proposes to distill the feature level knowledge from an additional teacher model to improve the NAS performance.

**NAS for ViT**   Most recently, several related works, e.g., AutoFormer (Chen et al., 2021a), and ViTAS (Su et al., 2021), have been proposed to search for ViTs. AutoFormer is the first paper that leverages NAS for ViT optimization. A comprehensive search space for the ViT architecture is proposed and a weight-entanglement training strategy is developed to improve the NAS efficiency. ViTAS leverages a similar supernet-based NAS method compared to AutoFormer and introduces private class token and self-attention maps to cater for the variance of distinct ViT architectures. Both work demonstrate promising accuracy improvement compared to the baseline DeiT models for large models with more than 1G FLOPs.

**Gradient cosine similarity in multi-mask learning**   A line of prior approaches have observed that the difficult training with multiple objectives can be improved by using the cosine similarity between gradients (e.g. Du et al., 2018; Yu et al., 2020b; Real et al., 2019). The cosine similarity is used to as a regularization or an indicator. Real et al. (2019) adds a regularization term to force the cosine similarity between two different losses to be larger than zero. Du et al. (2018) and Dery et al. (2021) propose to use gradient cosine similarity to identify whether auxiliary tasks can benefit the main task. In Du et al. (2018), once the cosine similarity is negative (gradient conflict), the weight of the auxiliary task is set to be zero. Yu et al. (2020b) is most related to our projection method, which projects the gradient of every loss to achieve orthogonal gradients. To avoid negative gradient cosine similarity, we project the gradient of the supernet to prioritize the training of sub-networks, which have similar intuition as auxiliary losses. In continual learning, many works uses orthogonal

gradient descent to restrict the direction of gradient updates of new tasks in order to avoid catastrophic forgetting (e.g. Farajtabar et al., 2020; Bennani et al., 2020; Saha et al., 2021).

# B SEARCH SPACE

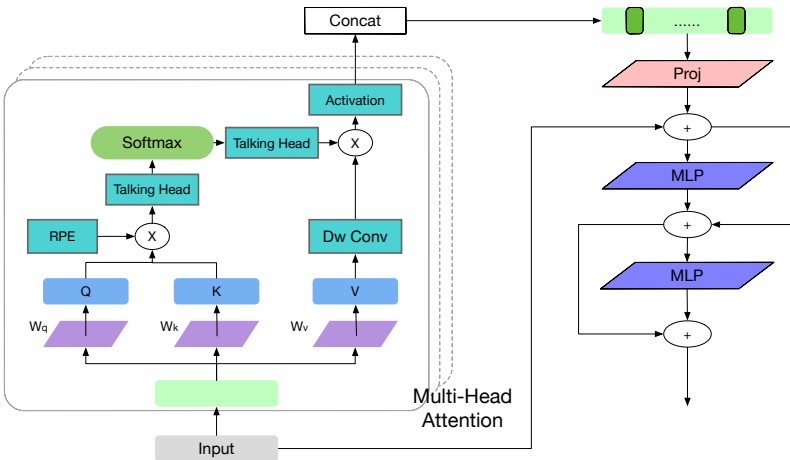

Figure 6: A demonstration of our self-attention module. 'RPE', 'Dw Conv, 'Talking Head', 'Proj' and 'MLP' refer to relative positional embedding, depth-wise convolutional layer, talking head attention, projection layer and MLP layer, respectively.

## B.1 EFFICIENT TRANSFORMER BUILDING LAYER

In this section, we give a detailed introduction about our transformer building layer with self-attention. In the literature, researchers have develop many variants of the standard self-attention with different focuses (e.g. efficiency, convergence, lone-term dependency, etc.). Motivated recent works, e.g., LeVit, SWIN-tranformer and VOLO (Graham et al., 2021; Liu et al., 2021; Yuan et al., 2021b), we develop a transformer layer for the purpose of efficiency and effectiveness in vision tasks. A demonstration of our transformer layer is shown in Figure 6. To enhance the learning capacity of our ViT models, we incorporate talking head (Shazeer et al., 2020) layers and depth-wise convolution layer in the self-attention module. Additionally, following LeViT (Graham et al., 2021), we expand the dimension of V matrix by expansion ratio 4 and introduce activation function after the projection matrix. Following Swin Transformer (Liu et al., 2021), we use relative positional embedding for the attention scores. For efficiency, we reduce the MLP expansion ratio to {1, 2} and add one additional MLP layer to keep the model complexity following MacaronNet (Lu et al., 2019).

**Positional information** The positional embedding in transformer architectures is location-dependent trainable parameters. Recent works propose absolute positional embedding, relative positional embedding or additional depth-wise convolution layers (Dong et al., 2021) to enhance the local information. We introduce two additional depth-wise convolution layers into a MHSA with relative positional embedding. For relative positional embedding, we directly follow the implementation in NLP (Shaw et al., 2018). For depth-wise convolution, we add one depth-wise convolution layer in the MLP layer and another depth-wise convolution layer after the linear transformations of V matrix.

**Expansion Ratio** In the self-attention design space, researchers have explored whether expanding the channels can have good performance. LeViT proposes to expand the dimension of V. We follow LeViT's design which expand the dimension of V by an expansion ratio 4. researchers have explored how many layers (Lu et al., 2019) should we use for MLP in a self-attention block. We follow the strategy developed by (Lu et al., 2019) which adds one more MLP layer for each self-attention block, but reduce the MLP expansion ratio to {1, 2} for efficiency. As displayed in Figure 6, we place an additional MLP layer after the first MLP layer.

**Normalization Layers and activation** Many recent works apply additional batch normalization layers, layer normalization layers or activation functions to the network. Taking the computation cost

of layer normalization layers into consideration, we do not introduce any new normalization layers to the basic self-attention layers.

**Talking-head attention and number of heads** Most of the existing ViTs set the dimension of each head to be 24/32 (e.g Touvron et al., 2020; 2021; Zhou et al., 2021; Wu et al., 2021; Liu et al., 2021). However, for a model with few channels, a large head dimension leads to few number of heads. We set a smaller head dimension (e.g. 8, 16) to make the number of heads to be large, and further introduce the talking-head attention to improve the capacity of different heads. Talking-head attention (Shazeer et al., 2020) introduce two additional linear transformation between all the heads, one is before softmax and another is after softmax.

**Classification head** Due to the use of depth-wise convolution layers and down-sampling, we remove the classification token for simplicity. While LeViT and DeiT use two heads for the teacher knowledge distillation and supervised labels, we use one head for all the training settings and replace the one-layer fully-connected layer head with MobilenetV3 (Howard et al., 2019) head so as to reducing computation cost.

**Scaling Factor** To train very deep transformer models, Touvron et al. (2021) introduces additional learnable channel-wise scaling factors initialized with $10^{-4}$ into the models. The channel-wise scaling factors are introduced to the output of each MLP and multi-head attention (MHA) layer in the model. Notice that many of sub-networks in our search space are very deep, we introduce architecture-dependent switchable scaling factors into the supernet (see Section 3).

## C    IMPLEMENTATION DETAILS ON TRAINING AND SEARCHING

### C.1    TRAINING AND SEARCH ALGORITHM

**Training**    Consider a supernet with trainable parameter $\theta$ and the candidate sub-networks set $\mathcal{A}$. The goal of training a supernet is to learn model parameter $\theta$ target at optimizing all the sub-networks in $\mathcal{A}$ and simultaneously achieving good accuracy. Let $s \sim \mathcal{A}$, $p(x;\theta)$ and $q(x;\theta_s)$ denote the output probability of the supernet and the sub-network $s$, we have the loss

$$\mathcal{L} = \ell(\theta) + \sum_{i=1}^{k} \ell_{\text{KD}}(\theta_{s_i}, \theta_{detach}), \ \ where \ \ \ell_{\text{KD}}(\theta_{s_i}, \theta_{detach}) = \mathbb{D}\bigg( p(x;\theta_{detach}) \ || \ q(x;\theta_{s_i}) \bigg).$$

Here, $\ell(\theta)$ denotes the loss of the supernet, $\theta_{s_i}$ represents the parameters for a sampled sub-network $s_i$, and $\mathbb{D}(p \ || \ q)$ is a divergence that measures the difference between $p$ and $q$. The $\theta_{detach}$ denotes a copy of $\theta$ whose gradient is stopped during back-propagation.

---
**Algorithm 1** Algorithm: Supernet based NAS training
---
1: **while** not converged **do**
2:       Sample a mini-batch data from dataset
3:       Sample the supernet (i.e., the biggest sub-network) from the search space and train the supernet with with ground truth labels (or with KD from an external teacher model)
4:       Sample $k$ random sub-networks from the search space and train them with KD by using the supernet network as the teacher model
5: **end while**

---

**Search**    After training, a random forest based neural predictor is trained to fit the map from the architecture hyper-parameters to the model performance (e.g., accuracy). A number of sub-networks are sampled from the trained supernet to train the neural predictor.

We then follow the strategy in previous works (e.g. Cai et al., 2019; Wang et al., 2021a) to do evolutionary search: 1) we randomly sample 1024 sub-networks from the supernet and estimate their accuracy on a sub-sampled subset of the ImageNet training set, which is never used during the supernet training; 2) we apply crossover and random mutation (see Zhou et al. (2011) for more details about evolutionary algorithms) on the best performing 128 sub-networks. We fix both the crossover size and mutation size to be 128, yielding 256 new sub-networks. We then evaluate the performance of these sub-networks; 3) We repeat the second step 40 times. The total number of sub-networks thus evaluated is around 10K.

---

**Algorithm 2** Algorithm: Supernet based NAS searching

---

1: **Input: a pretrained supernet with fixed weights**
2: Randomly sample 1024 sub-networks and evaluate their performance on a withhold training set (which is not used during training).
3: Partition 1024 sub-networks into training and validation subset with equal size. Train a random forest regressor to predict sub-netowrk's accuracy given the sub-network architecture hyper-parameters as the input.
4: Run evolution algorithm to search the Pareto of sub-networks. The sub-network accuracy is given by the random forest based predictor.

---

## C.2 ABLATION STUDIES

**Latency-aware Searching**   Note that the networks in the paper were optimized for the best FLOPs vs. accuracy trade-off. However, it is expected that the networks that achieve the best FLOPs vs. accuracy trade-off don't necessary yield the best latency vs. accuracy trade-off in the same time. Therefore, to achieve the best latency vs. accuracy trade-off, we re-search three NASViTs (B0/B1/B2) that form better latency vs. accuracy trade-off compared to the results from NASViTs (A0/A1/A2). For latency comparison, we evaluate the latency of NASViTs-A0/A1/A2 and AlphaNet-A0/A2/A4 on Intel(R) Xeon CPUs with a batch size of 1. See the result in the table below. Here, we do not remove BN or LN layers when estimating latency.

| Model | Accuracy (%) | CPU latency (ms) |
|---|---|---|
| NASViT-B0 | 78.2 | 21.0±0.4 |
| NASViT-A0 | 78.2 | 21.6±0.5 |
| AlphaNet-A0 | 77.9 | 21.4±0.5 |
| NASViT-B1 | 79.6 | 26.6±0.6 |
| NASViT-A1 | 79.7 | 27.2±0.6 |
| AlphaNet-A2 | 79.4 | 27.4±0.5 |
| NASViT-B2 | 80.6 | 29.1±0.6 |
| NASViT-A2 | 80.5 | 29.8±0.5 |
| AlphaNet-A4 | 80.4 | 30.4±0.5 |

Table 12: NASViT models searched with for better latency vs. accuracy trade-off.

# D   ARCHITECTURE VISUALIZATION OF NASViT MODELS

| | NASViT-A1 | NASViT-A2 | NASViT-A3 | NASViT-A4 |
|---|---|---|---|---|
| Conv | c: 16
d: 1
ks: 3
s: 2 | c: 16
d: 1
ks: 3
s: 2 | c: 16
d: 1
ks: 3
s: 2 | c: 16
d: 1
ks: 3
s: 2 |
| MBConv-1 | c: 16
d: 1
ks: 3
e: 3
s: 1 | c: 16
d: 1
ks: 3
e: 3
s: 1 | c: 16
d: 1
ks: 3
e: 3
s: 1 | c: 16
d: 1
ks: 3
e: 3
s: 1 |
| MBConv-2 | c: 24
d: 3
ks: 3
e: 4
s: 2 | c: 24
d: 3
ks: 3
e: 4
s: 2 | c: 24
d: 3
ks: 3
e: 5
s: 2 | c: 24
d: 3
ks: 3
e: 4
s: 2 |
| MBConv-3 | c: 32
d: 3
ks: 3
e:4
s: 2 | c: 32
d: 3
ks: 3
e: 6
s: 2 | c: 32
d: 3
ks: 3
e: 5
s: 2 | c: 32
d: 3
ks: 3
e:6
s: 2 |
| Transformer-4 | c: 64
d: 4
k: 1
e: 1
s: 2 | c: 64
d: 4
k: 1
e: 1
s: 2 | c: 64
d: 4
k: 1
e: 1
s: 2 | c: 64
d: 4
k: 1
e: 1
s: 2 |
| Transformer-5 | c: 112
d: 3
e: 1
s: 2 | c: 112
d: 3
e: 1
s: 2 | c: 112
d: 4
e: 1
s: 2 | c: 120
d: 3
e: 1
s: 2 |
| Transformer-6 | c: 160
d: 3
e: 1
s: 1 | c: 160
d: 5
e: 1
s: 1 | c: 160
d: 7
e: 1
s: 1 | c: 160
d: 6
e: 1
s: 1 |
| Transformer-7 | c: 216
d: 3
e: 1
s: 2 | c: 208
d: 4
e: 1
s: 2 | c: 216
d: 5
e: 1
s: 2 | c: 216
d: 6
e: 1
s: 2 |
| MBPool | c: 1792 | c: 1792 | c: 1984 | c: 1984 |
| Resolution | 192 | 224 | 256 | 288 |

Table 13: Here, 'c' denotes the number of output channels, 'd' denotes number of layers, 'ks' denotes kernel size, 'e' denotes expansion ratio, 'k' denotes number of windows, 's' denotes stride.

