# OpenReview forum: "NASViT: Neural Architecture Search for Efficient Vision Transformers with Gradient Conflict aware Supernet Training"
_ICLR.cc/2022/Conference — ICLR 2022 Poster_

### Official Review · Reviewer_oqFz · 2021-11-02

**Correctness:** 4
**Technical Novelty And Significance:** 4
**Empirical Novelty And Significance:** 3
**Recommendation:** 6
**Confidence:** 5

**Main Review:**

#Strength
1. The goal of introducing NAS to ViTs sounds reasonable.
2. The discovered gradient conflict issue looks interesting.
3.The experimental results seems solid.

#Weakness
1. It seems that the authors only focus on the one-shot NAS. Could other types of NAS methods be applied to ViTs? From this perspective, the discovered gradient conflict issue seems not universal.

2. The proposed three techniques, i.e., the gradient projection, the switchable layer scaling and the data augmentation, look like a “A+B+C” pattern. There seems to be no obvious connection among the three techniques.

3. Why do the authors remove the component of the super-net gradient that is conflict with the sub-network gradient? This operation is not intuitive. What are the negative effects of this operation?

4. The introducing of the weak data augmentation scheme is also sudden. It is highly suggested to give more intuitive explanations in Section 1.

5. What is the motivation of the raised three investigations?




**Summary Of The Paper:**

This paper aims at applying one-shot Neural Architecture Search (NAS) to Vision Transformers (ViTs). The authors claim that directly using existing CNN based NAS method to ViTs will lead to a gradient conflict issue. In order to tackle this issue, the authors propose three techniques, including a gradient projection, a switchable layer scaling, and a data augmentation. The experimental results demonstrate the effectiveness of the proposed method to some extent.

**Summary Of The Review:**

In general, the writing of this paper makes it difficult to follow the authors’ core thinking.

---

> ### Author Response · Authors · 2021-11-19
> **Official response to reviewer oqFz**
>
> We thank Reviewer #oqFz for your time and comments.
>
> *Q1. It seems that the authors only focus on the one-shot NAS. Could other types of NAS methods be applied to ViTs? From this perspective, the discovered gradient conflict issue seems not universal.*
>
> A. One-shot NAS achieves high search efficiency with significant reduction on GPU training costs and also demonstrates state-of-the-art results given various latency and compute constraints for various applications. Notable examples include BigNAS, One-for-all Networks, and AlphaNets, etc. Hence, we focus on developing accurate yet efficient ViT models with one-shot NAS.
>
> Weight-sharing based supernet training is one of the most important step for one-shot NAS as it directly determines the accuracy of all the sub-networks. Gradient conflict is the key problem we observed that impacts the ViT supernet convergence. Hence, resolving the gradient conflict problem to achieve the state-of-the-art accuracy vs compute trade-off becomes our focus. We also show the proposed method can be generally applied to training supernets for CNNs.
>
>
>
> *Q2: The proposed three techniques, i.e., the gradient projection, the switchable layer scaling and the data augmentation, look like a “A+B+C” pattern. There seems to be no obvious connection among the three techniques.*
>
> The three techniques are connected in that they are all designed to alleviate the gradient conflicting problem, from three different aspects: 1) optimizer (gradient projection)  2) architecture (switchable scaling layer)  and 3) training strategy (improved training recipe for ViT supernets). These three techniques are complementary to each other and can be combined together to improve the convergence of the ViT supernet.
>
> *Q3: Why do the authors remove the component of the super-net gradient that is conflict with the sub-network gradient? This operation is not intuitive. What are the negative effects of this operation?*
>
> A: When faced with gradient conflicts between the small and large networks, we need to prioritize the training for one of them. Intuitively, larger networks are easier to optimize and more robust to gradient noises due to over-parameterization. In contrast, smaller networks have limited learning capabilities and often require longer training schedule to reach a good performance. Since our target is develop small- and medium-sized ViT models, we choose to prioritize the training of small sub-networks by projecting the gradients of the largest sub-network. Empirically, we compare two strategies, i.e., prioritizing small networks (used in the paper) vs. prioritizing large networks in Table 7. As we can see, by prioritizing small networks, we can achieve higher accuracy for the small networks.
>
>
> *Q4: The introducing of the weak data augmentation scheme is also sudden. It is highly suggested to give more intuitive explanations in Section 1.*
>
> A: Using reduced data augmentation makes the training of the small sub-networks easier and hence reducing the gradient conflict. Intuitively, strong data augmentation is more likely to generate ambiguous examples that are very difficult to classify.  For these ambiguous examples, the small sub-networks might not have enough learning capabilities as the largest sub-network.
>
> In addition, applying strong regularization also increase the optimization difficulty for smaller sub-networks.
> As we validated in Table 7 (last row),  with "weak augmentation and regularization", the gradient conflict ratio is reduced from 34\% to 30\%, and in the meantime, yielding significant improvements on ImageNet for the smallest sub-network (0.8\% top-1 accuracy improvement as shown in Table 7).
>
> *Q5: What is the motivation of the raised three investigations?*
>
>
> A. The three investigations in section 2 are conducted to understand the inferior performance of standard supernet training method on ViT supernet.
> For one-shot NAS based on weight-sharing supernet, we believe there are 3 important aspects, including search space, training recipe, and supernet optimization. The first investigation is conducted from the search space design perspective, aiming to verify if the training bottleneck is caused by a bad search space design.
> The second investigation focuses on the training recipe, aiming to verify if other ViT training recipes would help improve close the performance gap.
> These two investigate confirms that our search space is well designed and naively applying prior-art ViT training recipes cannot produce good NAS results.
>
> The third investigation focuses on the supernet training optimization and provides an empirical evidence that
> the ViT supernet space tends to produce more gradient conflicts than standard CNNs.
> This motivates us to propose the techniques in Section 3 to resolve this issue, which allows us to achieve state-of-the-art accuracy vs compute trade-off as shown in section 4.

---

### Official Review · Reviewer_Y6a2 · 2021-11-02

**Correctness:** 3
**Technical Novelty And Significance:** 3
**Empirical Novelty And Significance:** 3
**Recommendation:** 5
**Confidence:** 5

**Main Review:**

Strengths:
1.	The motivation is clear, and the three proposed solutions are simple and generalizable.
2.	The proposed NASViT achieves promising performance in classification and segmentation with low computational cost.

Weakness:
1.	It is an interesting phenomenon that ViT supernet training suffers from the gradient conflict issue. But I am curious about the performance of other supernet training methods that don’t utilize KD from supernet, like once-for-all (ICLR20), DNANAS (CVPR20) and Cream (NeurIPS20).
2.	I am not clear why the authors introduce the switchable scaling. I don’t see the relationship between it and the gradient conflict issue, also the improvement in Table 7 looks marginal. Also Figure 3 is not very clear, e.g., what does c1 and c2 means?
3.	There is no experiment to support the statement “the supernet and the sub-networks are more likely to conflict with each other in the presence of stronger data augmentations and strong regularization.”. There is performance improvement shown in Table 7, but it is not clear whether this is caused by the conflict between supernet and sub-networks. And if so, I am wondering is there still an improvement for weak DA & Reg if the gradient projection is already applied.
4.	In Table 8, it seems that AlphaNets trained with alpha-divergence doesn’t benefit from your method. Maybe you should show the gradient conflicts ratio for it to give some insights.
5.    Since the gradient projection leads to slow convergence, the authors should provide the training time cost before and after using the proposed gradient projection, as the training time cost is also important.
6.	How about the training time consuming after using the other two techniques.  Does it reduce the training time caused by gradient projection for its slow convergence? Detailed information should be presented.
7.	Why do the authors use EfficientNet-B5 as the teacher? Why not B6，B7 or other networks? The reasons are not mentioned in this paper. It is confusing.
8.    To be more general and convincing, though the evaluations of the smallest and the largest networks are provided, I think the results of the middle-size network in needed experiments are also critical. For example, in Fig 2, Tab 3, and so on.
9.    The authors only adopt gradient projection on CNN-based supernet while conducting ablation studies. The generalizability of the other two techniques in CNN should be verified as well like Tab.7.




**Summary Of The Paper:**

This work presents the gradient conflict issue in ViT training, i.e., the gradients of sub-networks conflict with that of the supernet, leads to inferior performance of ViT supernet training. The authors fix this issue by 1) a gradient project method to prioritize the sub-network update; 2) Use switchable layers to increase the model capacities of sub-networks; 3) Simplify the training recipe. The proposed NASViT shows the state-of-the-art top-1 accuracy v.s. FLOPs trade-offs on ImageNet.

**Summary Of The Review:**

The motivation of this paper is not very convincing, need to provide more evidence to prove. For more details, pls see the weaknesses.
Also, there are some recent works on NAS+ViT missing in the related works, such as "CATE: Computation-aware Neural Architecture Encoding with Transformers", "Vision Transformer Architecture Search" and "AutoFormer: Searching Transformers for Visual Recognition". It is suggested to add them into comparison and discussion, especially for Autoformer which might be the first ViT+NAS work and has been published in ICCV21. I will re-consider the rate if all the concerns are well addressed or not.

---

> ### Author Response · Authors · 2021-11-19
> **Official response to reviewer Y6a2**
>
> We thank Reviewer #Y6a2 for your time and comments.
>
> *Q1: It is an interesting phenomenon that ViT supernet training suffers from the gradient conflict issue. But I am curious about the performance of other supernet training methods that don’t utilize KD from supernet, like once-for-all (ICLR20), DNANAS (CVPR20) and Cream (NeurIPS20).*
>
> A: Thanks for the great suggestions on the supernet training algorithms. In this work, we follow the supernet-based NAS practices proposed in BigNAS and AlphaNet as they have demonstrated SOTA accuracy and compute trade-off. We have tried using supernet for in-place KD and using pre-trained teacher (EfficientNet-B5) for regular KD, both of which suffer from the gradient conflict issue. For other algorithms, including once-for-all (progressive shrinking), DNA NAS (block-wise distillation from an additional teacher) and Cream (using a prioritized path sub-network as the teacher), we added the citation to these papers in our updated manuscript but did not try these methods during our exploration. We regard further study on block-wise distillation and progressive shrinking as our future direction.
>
> *Q2: why the authors introduce the switchable scaling. I don’t see the relationship between it and the gradient conflict issue, also the improvement in Table 7 looks marginal. Also Figure 3 is not very clear, e.g., what does c1 and c2 means?*
>
> A: The proposed switchable scaling layer is motivated by the switchable batch normalization layer in Slimmable Neural Network (ICLR 2019). The basic idea is to unshare part of the network, which can be updated freely, to reduce the gradient conflicts between the supernet and the sub-networks. Intuitively, the more parameters are unshared, the better the gradient conflicts can be resolved. However, parameter unsharing can increase the parameter size for the supernet and lead to extra memory usage. Hence, we want to unshare a small set of parameters to alleviate gradient conflicts without adding too many extra parameters. Empirically, we find the switchable scaling layer helps us achieve the goal.
> "c1" and "c2" in Figure 3 represent the number of channels activated in a self-attention layer and MLP, respectively. For the switchable scaling layer, we have a different scaling factor for each choice of MLP or self-attention width. For example, in the search space in Table 1, for Transformer-4, there are 2 different choices (i.e., {64, 72}) for the widths. Then, the switchable scaling layer has 2 sets of scaling factors.
>
> *Q3: There is no experiment to support the statement “the supernet and the sub-networks are more likely to conflict with each other in the presence of stronger data augmentations and strong regularization.”. There is performance improvement shown in Table 7, but it is not clear whether this is caused by the conflict between supernet and sub-networks. And if so, I am wondering is there still an improvement for weak DA \& Reg if the gradient projection is already applied.*
>
> A: In the last row of Table 7, we show the gradient conflict ratio could be reduced from 34\% to 30\% by switching from strong data augmentation and regularization to weak data augmentation and regularization. This results validate our argument -  “the supernet and the sub-networks are more likely to conflict with each other in the presence of stronger data augmentations and strong regularization."
> In the following, we provide additional studies to understand whether "weak DA \& Reg" is necessary when "gradient projection" and "switchable scaling layer" are applied. To this goal, we finetune our final ViT-supernet checkpoint for 10 epochs with typical strong data augmentation like AutoAugment, RandAugment and Cutmix, respectively, and summarize our findings in the below.
>
> As we can see from the table, slightly fine-tuning our best model with strong data augmentation will lead to a noticeable performance degradation for the smallest sub-network. Our proposed training strategy with weak data augmentation yield the best results.
>
> | | AutoAugment | RandAugment | CutMix | Smallest sub-network | Largest sub-network |
> | --- | --- | -- | --- | --- | --- |
> | Default | x | x | x | 78.1  | 82.9 |
> | Setting 1 | ✓ | x | x | 77.9 | 82.8 |
> | Setting 2 | x | ✓ | x | 77.8 | 82.8 |
> | Setting 3 | x | x | ✓ | 77.8 | 82.8 |
>
> Table: Fine-tuning results with different data augmentation schemes on ImageNet. All supernets are trained with "gradient projection" and "Switchable scaling layer". RandAugment refers to the standard implementation with n=2 and m=9 (n: number of augmentation transformations, m: magnitude).

---

> > ### Author Response · Authors · 2021-11-19
> > **Official response to reviewer Y6a2**
> >
> >
> > *Q4: In Table 8, it seems that AlphaNets trained with alpha-divergence don't benefit from your method. Maybe you should show the gradient conflicts ratio for it to give some insights.*
> >
> > A: The gradient conflicts ratio from AlphaNets trained with alpha-divergence is presented in Table 4. In the paragraph `Improving CNN-based supernets', we discussed the connection with Table 4. We kindly refer Reviewer \#Y6a2 to this paragraph for more details.
> >
> > *Q5/Q6: Since the gradient projection leads to slow convergence, the authors should provide the training time cost before and after using the proposed gradient projection, as the training time cost is also important. How about the training time consuming after using the other two techniques. Does it reduce the training time caused by gradient projection for its slow convergence? Detailed information should be presented.*
> >
> > A: First, we would like to clarify that our gradient projection technique does NOT lead to a slow convergence.
> > Instead, taking advantage of non-conflicting gradients, our technique can achieve the same accuracy with fewer number of epochs. For examples, with our techniques, the smallest and largest sub-network already achieves 77.7\% and 82.6\% top-1 accuracy, respectively, after just 100 epochs of training.  Without our techniques, the performance of the smallest and largest sub-network would saturate at 76.6\% and 82.2\%, respectively, as shown in Figure 2.
> > A more important benefit of our gradient projection technique is the accuracy gain, which indicates a much better convergence of the supernet and can be applied to both ViT supernet (Table 7) and CNN supernet (Table 8).
> >
> > *Q7: Why do the authors use EfficientNet-B5 as the teacher? Why not B6, B7 or other networks. The reasons are not mentioned in this paper. It is confusing.*
> >
> > A: As we explained in Section 4.1, our baseline FBNet-v3 uses a RegNetY-32G teacher with 84.5\% top-1 accuracy and LeViT uses a RegNetY-16G teacher with 83.6\% top-1 accuracy. To have a fair comparison with these baselines, we pick a similarly accurate teacher model, a Efficient-B5 with 83.3\% top-1 accuracy, to clearly demonstrate the benefits from our techniques without taking advantage of a stronger teacher.
> >
> > *Q8: To be more general and convincing, though the evaluations of the smallest and the largest networks are provided, I think the results of the middle-size network in needed experiments are also critical. For example, in Fig 2, Tab 3, and so on.*
> >
> > A: As shown in Slimmable Networks, BigNas and AlphaNets, the smallest and the largest sub-network often form the lower and the upper bound performance, respectively, providing a good indication of the overall NAS performance.
> >
> > Below we show some additional results on middle size networks. We follow our settings in Table 3 and provide additional results for NASViT-A1, NASViT-A3 in the below. The observations from NASViT-A1 and NASViT-A3 are consistent with the trends observed from the smallest and the largest sub-network and hence, confirming the sufficiency of our results.
> >
> > | | FLOPs | AlphaNet | DeiT | LeViT |
> > | --- | --- | --- |--- |--- |
> > | Smallest | 208M | 77.0 | 76.6 | 76.8 |
> > | NASViT-A1 | 309M | 78.3 | 78.1 | 78.3 |
> > | NASViT-A3 | 528M | 79.9 | 79.7 | 79.7 |
> > | Largest | 1.9G | 82.4 | 82.2 | 82.2 |
> >
> >
> > *Q9: The authors only adopt gradient projection on CNN-based supernet while conducting ablation studies. The generalizability of the other two techniques in CNN should be verified as well like Tab.7.*
> >
> > A:
> > In Table 7, we conduct the ablation for the contribution of each technique. In Table 8, the goal is to verify the generalizability of our three techniques to the CNN supernets. In this setting, we applied all three techniques together to improve CNN-based supernets. We will remove the sentence, "we adapt the AlphaNet training to prioritize the sub-networks update in case of gradient conflicts", to improve clarity.
> >
> > *Q10: Related works.*
> >
> > A: Thanks for pointing out these interest papers. We have already added them in our related work in our revision.
> > As we show in the following table, compared to ViTAS, CATE and AutoFormer, our NASViTs still show clear empirical advantage.
> >
> > | Architecture | FLOPs | Top-1 Accuracy |
> > | --- | --- | --- |
> > | AutoFormer | 0.6G | 75.0\% |
> > | ViTAS-C  | 1.0G | 74.7\% |
> > | AutoFormer | 1.3G | 74.7\% |
> > | ViTAS-D   | 1.6G | 76.2\% |
> > | **NASViT-A0** | **0.2G** | **78.2\%** |
> >
> > Table: Comparison with ViT NAS methods.

---

> > > ### Comment · Reviewer_Y6a2 · 2021-11-22
> > > **Thanks for the response**
> > >
> > > Thanks for the responses and detailed revisions, which addressed most of the concerns. However, the discussion with the most related works should be added into the main manuscript, instead of Appendix A. I understand the space is limited, but for a research paper, the discussion with the most related literature is very important and shall be seriously considered and well addressed. Also, pls move the comparisons into Tab. 6, rather than just putting them here for review.
> > >
> > > Some claims are not correct. For example, "Compared to these work, we focus on a different NAS paragradigm (i.e., the supernet-based NAS)". Actually, the previous works AutoFormer and ViTAS are also supernet-based NAS.
> > >
> > > paragradigm (a typo in the paper.)

---

> > > > ### Author Response · Authors · 2021-11-23
> > > > **Thanks for further clarification and advice**
> > > >
> > > > Thanks for your suggestions. We re-summarize these works, and have added an additional paragraph in the introduction section to discuss the most related works and refer the readers to the appendix for more related works.

---

### Official Review · Reviewer_XjJo · 2021-11-02

**Correctness:** 3
**Technical Novelty And Significance:** 3
**Empirical Novelty And Significance:** 3
**Recommendation:** 6
**Confidence:** 3

**Main Review:**

+ The writing is easy to follow and read.
+ The evaluation results are promising.
-  The motivation of this work is clear, but the explanation of poor performance (gradient conflict issue) is somewhat  not convincible.
-  The architecture in the search space consists of MBConv and transformer, it is better to compare with the existing works that also combine the CNN and transformers, like PiT, etc.


**Summary Of The Paper:**

To solve the gradient conflict issue that affect the performance，this work propose a series of techniques, including a gradient projection algorithm, a switchable layer scaling design, and a simplified data augmentation and regularization training recipe to improve the convergence and the performance of all sub-networks.

**Summary Of The Review:**

It is a good work with clear motivation and  promising performance, the reason of poor performance by directly applying the supernet-based NAS to optimize ViTs need to be discussed more.

---

> ### Author Response · Authors · 2021-11-19
> **Reply to Reviewer XjJo**
>
> We thank Reviewer #XjJo for your time and comments.
>
> *Q1: The motivation of this work is clear, but the explanation of poor performance (gradient conflict issue) is somewhat not convincing.*
>
> A: To our understanding, the question is about
> the connections between the poor performance and gradient conflicts. After observing the poor performance, we conduct investigations in Section 2 to understand its origin from three different directions, including search space, training recipe, and optimization. The first and second investigations confirm that our search space is well designed and directly applying existing prior-art ViT training recipes cannot help improve the NAS results.  The third investigation focuses on the supernet training optimization and provides an empirical evidence that the ViT supernet space tends to produce more gradient conflicts than standard CNNs.
> This motivates us to "hypothesize that the inferior performance of our ViT supernet is mainly caused by the large percentage of disagreements between the supernet gradients and the sub-networks gradients." and propose the techniques in Section 3 to resolve this issue.
>
> How effective each of the proposed techniques is in reducing the gradient conflict issue and consequently improving the NAS performance?
> For this matter, we would like to refer Reviewer \#xjJo to Table 7 for our detailed ablation studies.
> And as we demonstrated in Table 7,
> both "weak DA \& Reg" and "Switchable scaling" would be able to effectively reduce the gradient conflict ratio and in the meantime, leading to improved NAS performance on ImageNet.
> Additionally, "Prioritize (sub)" will eliminate the conflicted gradient components from the supernet and yield the best performance.
>
> *Q2: The architecture in the search space consists of MBConv and transformer, it is better to compare with the existing works that also combine the CNN and transformers, like PiT, etc.*
>
> A: In this work, we compare  with hybrid models like LeViTs. To our best knowledge, LeViT models are the state-of-the-art hybrid network that achieves quite competitive performance on small- and medium-sized network (<600M FLOPs). As shown in both Table 6 and Figure 4, our NASViTs outperform LeViTs by a large margin.
>
> In addition, thank you for pointing out the interesting reference on PiT. We have revised our draft to include a discussion on PiT. Compared with PiT, our NASViTs still has clear empirical advantage as we show in the following table.
>
> | Architecture | FLOPs | Accuracy |
> | --- | ---- | ---- |
> | PiT-Ti | 0.7B | 74.6\% |
> | PiT-XS | 1.4B | 79.1\% |
> | NASViT-A5 | 0.7B | 81.8\% |
> Table: comparisons with PiT.

---

### Official Review · Reviewer_G3NT · 2021-11-02

**Correctness:** 3
**Technical Novelty And Significance:** 3
**Empirical Novelty And Significance:** 3
**Recommendation:** 8
**Confidence:** 4

**Main Review:**

I really like the exposition of the paper, going through hypotheses and presenting experiments supporting or disproving the hypotheses. This was great to read. I also really appreciate the links to literature in different fields, speicifically multi-task learning regarding conflicting gradients.

1. My main concern is that the paper is about finding efficient and well-performing models, however it does not include a single timing anywhere I could see. I would really appreciate if Table6 included a latency column, and there was a variant of Figure4 with latency as x-axis. Especially the small attention head size has me concerned, whether the good FLOPs will actually translate to good efficiency.
2. What about gradient agreement between individual random subnetworks? Is it better, equal, or worse than the super network vs average f sub networks?

**Summary Of The Paper:**

The paper investigates applying a SOTA NAS method (from AlphaNet) to a SOTA ViT search-space (from LeViT) and notices disappointing results. Following an investigation, conflicting gradients are identified as the root cause, alleviated via three components: gradient projection, learnable scaling factors, and heavily reduced augmentations.

**Summary Of The Review:**

I find the paper generally of high quality, but the lack of any timings is concerning. If this can be fixed, it would be a pretty good paper.

---

> ### Author Response · Authors · 2021-11-19
> **Reply to Reviewer G3NT**
>
> We thank Reviewer \#G3NT for your time and comments.
>
> *Q1: My main concern is that the paper is about finding efficient and well-performing models, however, it does not include a single timing anywhere I could see. I would really appreciate if Table 6 included a latency column, and there was a variant of Figure 4 with latency as x-axis. Especially the small attention head size has me concerned, whether the good FLOPs will actually translate to good efficiency.*
>
> A: Thank you for your advice.
> In the paper, to enable a fair comparison with prior art baselines like AlphaNet, FBNetV3, etc, we follow the same evaluation protocols used in these baselines and designed/optimized our models for the best FLOPs vs accuracy trade-offs.
> For latency comparison, we evaluate the latency of NASViTs-A0/A1/A2 on Intel(R) Xeon CPUs with a batch size of 1.
> Compared to AlphaNets, our NasViTs have similar CPU latency while achieving better accuracy.
> Please see the comparison below:
>
> | | AlphaNet | NASViT |
> | --- | --- |--- |
> | | A0 | A0 |
> |Accuracy (\%)| 77.9 | 78.2 |
> | CPU (ms )| 21.4$\pm$0.5 | 21.6$\pm$0.5 |
> | | A2 | A1  |
> |Accuracy (\%)| 79.4 | 79.7 |
> | CPU (ms )| 27.4$\pm$0.5 | 27.2$\pm$0.6 |
> | | A4 | A2  |
> |Accuracy (\%)| 80.4 | 80.5 |
> | CPU (ms )| 30.4$\pm$0.7 | 29.8$\pm$0.5 |
>
> Note that the networks in the paper were optimized for the best FLOPs vs. accuracy trade-off. However, it is expected that the networks that achieve the best FLOPs vs. accuracy Pareto don't necessary yield the best latency vs. accuracy Pareto in the same time. Therefore, to optimize the best latency vs. accuracy trade-offs, we re-search three NASViTs (B0/B1/B2) that form better latency vs. accuracy trade-offs compared to the results from NASViTs (A0/A1/A2). See the result in the table below.
>
> | Model | Accuracy (\%) | CPU latency (ms) |
> | --- |--- |--- |
> | B0   | 78.2 |  **21.0$\pm$0.4** |
> | A0   | 78.2 |  **21.6$\pm$0.5** |
> | B1   | 79.6 |  **26.6$\pm$0.6** |
> | A1   | 79.7 |  **27.2$\pm$0.6** |
> | B2   | 80.6 |  **29.1$\pm$0.6** |
> | A2   | 80.5 |  **29.8$\pm$0.5** |
>
> Table: NASViT models searched with for better latency.
>
> *Q2: What about gradient agreement between individual random subnetworks? Is it better, equal, or worse than the super network vs average f sub networks?*
>
>
> A: In the following, we follow your suggestion to compare the gradient conflict between 1) random sub-network vs. random sub-network, and 2) random sub-network vs. supernet. Specifically, we follow our settings in Table 4 and pick the last checkpoint (epoch 360th) for evaluation. As shown in the comparison below, the gradient conflicts between the random sub-network and the supernet is larger.
>
> The comparison is also aligned with our intuition that the gradient conflict between the supernet and the small sub-networks is more important. This is because in our method, small sub-networks (e.g., the smallest sub-network and random sub-networks) are supervised by the soft labels generated by the largest sub-network (i.e., the supernet) with knowledge distillation loss while the supernet is supervised with cross-entropy loss (or knowledge distillation loss with an EfficientNet-B5 teacher). The difference in loss functions, as well as the network capacities, also tend to increase the gradient conflict between the supernet and the sub-networks.
>
> | | random | supernet |
> | --- | --- |  --- |
> |  Negative Cosine Similarity Ratio | 26\% | 31\% |
>
> Table: Gradient conflict ratios for random sub-networks, with the same settings in Table 4.

---

> > ### Comment · Reviewer_G3NT · 2021-11-21
> > **Thank you**
> >
> > I thank the authors for their additional experiments regarding timing, and the verification of random subnet gradient disagreement. I believe if the authors manage to squeeze these results into the paper (in the worst case, in an appendix), it will make a reader like me even more convinced of this already good paper.

---

### Official Review · Reviewer_GnU6 · 2021-11-03

**Correctness:** 3
**Technical Novelty And Significance:** 4
**Empirical Novelty And Significance:** 3
**Recommendation:** 6
**Confidence:** 4

**Main Review:**

## Strengths:

1. The paper does not only transfer the NAS algorithms from CNN to transformer, but also observes the gradient conflict dilamma and resolves it partly.

2. The searched architecture seems over-performs swin, cvt, and some other vision transformers.

3. Besides gradient conflict, the authors also improve the performance by changing data augmentation.

## Weaknesses & Questions:

1. I am not sure what "Group" in Table 6 means. Does it mean the number of parameters? As I know, most vision transformers have parameters less than 100M.

2. The improvement on segmentation tasks seems a bit marginal.

3. It is recommended to explore how other data augmentation methods affect the NAS performance, e.g., AutoAugment, color jitter, cropping, etc. All these are common data augmentation methods for vision transformers.

**Summary Of The Paper:**

The paper proposes a NAS algorithm for vision transformers. The search space contains both CNN blocks and transformer blocks. The authors solve the gradient conflict dilemma in this process and find out  a good architecture. Experiments on classification and segmentation show its effectiveness over other vison transformers like swin, cvt, etc.

**Summary Of The Review:**

Though there are still some minor flaws in the paper, I think it is a good one and is nearly ready for publication.

---

> ### Author Response · Authors · 2021-11-19
> **Reply to Reviewer GnU6**
>
> We thank Reviewer \#GnU6 for your time and comments.
>
> *Q1: I am not sure what "Group" in Table 6 means. Does it mean the number of parameters? As I know, most vision transformers have parameters less than 100M.*
>
> A: In table 6, we group different models according to their computation cost (i.e., FLOPs). For example, "300 - 400M" indicates that the models in these rows all have computation cost in the range between 300 MFLOPs and 400 MFLOPs. This comparison follows existing papers including FBNet, AlphaNet, LeViT, BigNas and Once-for-all.
> Our NASViT models have relatively small parameter sizes (e.g., <10M number of parameters).
>
> *Q2: The improvement on segmentation tasks seems a bit marginal.*
>
> A: To demonstrate the transfer learning ability of the developed NASViTs on segmentation tasks, we compare our method with strong baselines like MobileNetV3, AlphaNet and SegFormer. In particular, AlphaNets are CNN backbones that achieve the state-of-the-art (SOTA) classification accuracy vs. compute (FLOPs) trade-off on Imagenet. SegFormers are specifically designed for semantic segmentation and achieve SOTA segmentation accuracy vs. compute (FLOPs).
>
> Compared to these strong prior-art baselines, our NASViT models deliver clear gains on both CityScapes and ADE20K, achieving a better FLOPs vs. mIoU Pareto front. For example, on CityScapes and ADE20K, NASViTs outperform AlphaNets with an average of 1.4 and 0.8 mIoU, respectively. And NASViTs outperform SegFormer models with an average of 1.7 and 0.5 mIoU on CityScapes and ADE20K, respectively.
>
> *Q3: It is recommended to explore how other data augmentation methods affect the NAS performance, e.g., AutoAugment, color jitter, cropping, etc. All these are common data augmentation methods for vision transformers.*
>
> A:  Thank you for your suggestion. In Table 3, we experimented with the different training recipes suggested in LeViT, DeiT, and AlphaNet. Specifically,
>
> - LeViT uses a combination of mixup, cutmix, repeated augmentation and random erasing for data augmentation
>
> - DeiT uses a combination of mixup, cutmix, RandAugmentation, repeated augmentation and random erasing for data augmentation.
>
> - AlphaNet uses a combination of AutoAugment and label smoothing for data augmentation.
>
> We copy Table 3 in the paper below for the convenience of the reviewer and add the comparison to our training recipe below. As shown in the table, these advanced data augmentation strategies do not help with the ViT supernet training.
>
> | | Training Recipes | AlphaNet | DeiT | LeViT | Ours |
> | --- | --- | --- |--- |--- |--- |
> | Smallest | 208M | 77.0 | 76.6 | 76.8 | 77.4 |
> | Largest | 1.9G | 82.4 | 82.2 | 82.2 | 82.5 |

---

> > ### Comment · Reviewer_GnU6 · 2021-11-29
> > **Thanks for your responses**
> >
> > Thanks for the authors' responses which have resolved my concerns. I think it would be better if the authors can supplement the explaination about "Group" in Table 6's caption in their final revision.

---

> > > ### Author Response · Authors · 2021-11-29
> > > **Thank you**
> > >
> > > Reviewer GnU6, thanks for your comments. We will add the explanation about 'groups' in the caption!

---

### Decision · Program_Chairs · 2022-01-20

**Decision:**

Accept (Poster)

**Comment:**

This paper tackles a very timely problem.
Scores of 5,6,6,8 put it in the borderline region, but in the private discussion the more negative reviewer noted that they would also be OK with the paper being accepted. I therefore recommend acceptance.
Going through the paper I missed any mention of available source code. I strongly recommend that the authors make code available; this would greatly increase the paper's impact.